# Protocol for a Case Control Study to Evaluate Oral Health as a Biomarker of Child Exposure to Adverse Psychosocial Experiences

**DOI:** 10.3390/ijerph19063403

**Published:** 2022-03-14

**Authors:** Anna Durbin, Bennett T. Amaechi, Stephen Abrams, Andreas Mandelis, Sara Werb, Benjamin Roebuck, Janet Durbin, Ri Wang, Maryam Daneshvarfard, Konesh Sivagurunathan, Laurent Bozec

**Affiliations:** 1MAP Centre for Urban Health Solutions, Unity Health Toronto, Toronto, ON M5B 1W8, Canada; ri.wang@unityhealth.to (R.W.); maryam.danesh@unityhealth.to (M.D.); 2Department of Psychiatry, University of Toronto, Toronto, ON M5T 1R8, Canada; 3Department of Comprehensive Dentistry, University of Texas Health Science Center at San Antonio, San Antonio, TX 78229, USA; amaechi@uthscsa.edu; 4Cliffcrest Dental Office, Four Cell Consulting, Quantum Dental Technologies, Toronto, ON M6B 1L3, Canada; dr.abrams4cell@sympatico.ca; 5Center for Diffusion-Wave and Photoacoustic Technologies (CADIPT), University of Toronto, Toronto, ON M5T 1R8, Canada; mandelis@mie.utoronto.ca (A.M.); konesh@thecanarysystem.com (K.S.); 6Toronto Children’s Dentistry, Toronto, ON M5T 1R8, Canada; sarabwerb@gmail.com; 7Victimology Research Centre, Algonquin College, Ottawa, ON K2G 1V8, Canada; roebucb1@algonquincollege.com; 8Provincial System Support Program (PSSP), Centre for Addiction and Mental Health (CAMH), Toronto, ON M5S 2S1, Canada; janet.durbin@camh.ca; 9Faculty of Dentistry, University of Toronto, Toronto, ON M5G 1G6, Canada; l.bozec@dentistry.utoronto.ca

**Keywords:** enamel anomalies, resilience, adversity, childhood, teeth, developmental defects of enamel, visual assessment, photothermal radiometry and modulated luminescence, truncated correlation-photothermal coherence tomography, microcomputed tomography, white spot lesions

## Abstract

Background: The early identification of children who have experienced adversity is critical for the timely delivery of interventions to improve coping and reduce negative consequences. Self-report is the usual practice for identifying children with exposure to adversity. However, physiological characteristics that signal the presence of disease or other exposures may provide a more objective identification strategy. This protocol describes a case–control study that assesses whether exposure to adversity is more common in children with tooth enamel anomalies compared to children without such anomalies. Methods: For 150 mother–child pairs from a pediatric dental clinic in Toronto, Canada, maternal interviews will assess the child’s adverse and resilience-building experiences. Per child, one (exfoliated or extracted) tooth will be assessed for suspected enamel anomalies. If anomalies are present, the child is a case, and if absent, the child is a control. Tooth assessment modalities will include usual practice for dental exams (visual assessment) and modalities with greater sensitivity to identify anomalies. Conclusion: If structural changes in children’s teeth are associated with exposure to adversity, routine dental exams could provide an opportunity to screen children for experiences of adversity. Affected children could be referred for follow-up.

## 1. Introduction

In North America and internationally, it is estimated that nearly two-thirds of children experience significant adversity before age 18 [1,2], including physical and emotional abuse, neglect, hardships, and exploitation.

Experiencing childhood adversity can cause prolonged activation of the body’s stress responses. In the absence of protection and support, this can produce “toxic stress”. Toxic stress can contribute to adverse outcomes over the course of a person’s life [3,4], including developmental disruptions; poor social functioning; low educational attainment; high-risk behaviors (e.g., suicide attempts, sexual risk taking, alcohol and drug misuse, anti-social behavior, violence); physical health conditions (e.g., diabetes, obesity, cardiovascular disease, respiratory disease, liver disease, chronic lung disease); mental health conditions (e.g., psychosis, depression, anxiety/posttraumatic stress disorder) [5]; and shorter life expectancies; sometimes by as much as 20 years. These adverse outcomes are more common and more severe for people who have experienced multiple childhood adversities, especially four or more (dose–response relationship) [1,2,3,4,6,7,8,9,10]. The costs associated with these adverse outcomes are enormous—estimated to be USD 748 billion annually in North America (3.6% of GDP) and USD 581 billion annually in Europe (2.7% of GDP) [11].

Not all exposures to childhood adversity result in negative consequences. The presence of resilience or an ability to adapt well in the face of stressful situations can reduce the impact of the traumas and subsequent negative outcomes [12,13]. For children exposed to adversity, greater resilience is associated with lower rates of physical and mental health issues [14,15,16,17,18,19,20,21,22] and behavior problems throughout development [23]. Interventions are available to enhance resilience, but have the greatest potential to achieve benefit and cost-savings if provided early [24,25]. As such, strategies to reliably identify children with early life adversity are needed [26]. Self-report is the primary strategy for identifying exposures to adversity, but has been criticized for poor accuracy due to factors such as social desirability [27,28], recall limitations, and underreporting [29,30].

Given these challenges, there is a need for strategies and measures beyond self-report that can more reliably flag children’s exposure to adversity and direct interventions [27,31]. Biomarkers (short for biological markers) are physiological characteristics that can serve as signs of disease or other exposures [32]. Biomarkers may serve this purpose if they are identifiable with limited cost and minimally invasive approaches [27,31]. For example, changes in the brain physiology may occur after experiences of adversity [32,33,34,35], but routine neuroimaging of every child’s brain to support early identification is not feasible.


*Tooth enamel anomalies as biomarkers of childhood adversity*


Tooth enamel is formed and mineralized in daily increments during the course of tooth development, which occurs on and off from birth until the mid-teenage years. Severe disturbances in the enamel manifest as clinically detectable defects which may occur on different tooth surfaces, in different forms, and may be related or independent from each other. Enamel anomalies present clinically as opaque white sections on teeth due to porous and irregular microstructures that scatter light [36]. These anomalies can be seen on the enamel surface and/or in its ultrastructure, and can affect the whole thickness of enamel or a localised area within the enamel [37].

Emerging evidence suggests that exposure to stress may disrupt the formation of tooth enamel [27,35,38]. In studies of gorillas, chimpanzees and orangutans, tooth enamel anomalies have been shown to develop during periods when the primate felt stress, for example, due to social disruptions, disease, and physical trauma [39,40,41,42,43,44,45,46]. These defects become “permanent biochemical signatures” of experiences of stress on primate teeth [46], p. 6.

In humans, a small number of recent studies support the association between childhood adversity and elements of poor oral health. One study reported that compared to people with no adverse childhood events (ACEs), people with four or more ACEs were 2.8 times more likely to have dental problems (not defined), although the methods were not cited [47]. Another study analyzed data from the 2011–2012 National Survey for Child Health (NSCH). It showed that relative to children who had experienced no ACEs, children who had experienced three or more ACEs were more likely to have teeth in fair or poor condition and to have experienced toothaches, decayed teeth and/or unfilled cavities [48].

A recently published study [31] examined the association between prenatal and perinatal maternal psychosocial factors and the width of neonatal lines in the canine teeth of 70 children. Neonatal lines were wider in the teeth of children born to mothers who self-reported severe lifetime depression, or any lifetime psychiatric problems. Similarly, greater anxiety or depressive symptoms at 32 weeks of gestation were associated with wider lines. However, neonatal lines were narrower in children whose mother reported high social support after birth.

To the author’s knowledge, only one study has examined the relationship between stressful life events and enamel anomalies in children [49]. Children who experienced changes of address, hospitalizations, accidents/falls, illnesses, medication use, and weight loss were more likely to have enamel anomalies in their permanent incisor teeth than children who had not experienced these events.

One limitation of these studies was the sensitivity of the modality for assessing tooth health, which may have limited their ability to assess tooth health as a potential biomarker of childhood adversity exposure. Another is that the impact of exposures to other childhood adversities such as abuse, neglect, and hardship were not examined.

In summary, dental enamel anomalies hold promise as a biomarker of child exposure to adversity that can facilitate early identification. However, further investigation is needed to assess whether the association exists and whether the identification of anomalies during routine dental exams is sensitive enough to be used for this purpose. If the association is present, there may be a role for oral health care professionals (OHCPs) in screening and referring affected children to supports. In addition to being a more objective screening strategy than self-report, some children see dentists more regularly than physicians [50].


*Study Aims*


This protocol describes a study to assess the relationship between exposure to adversity in children and the presence of dental enamel anomalies as identified in clinical practice through standardized visual assessment (primary outcome). The secondary aim is to assess the relationship between exposure to adversity in children and the presence of dental enamel anomalies as assessed using three more sensitive methods not currently available for routine clinical use—photothermal radiometry and modulated luminescence (PTR-LUM), truncated correlation-photothermal coherence tomography (TC-PCT), and microcomputed tomography (secondary outcomes). These additional modalities will provide a more sensitive test of the presence of the association as a proof-of-concept.

In each analysis, the role of resilience in mediating the relationship between childhood adversities and enamel anomalies will also be examined.

## 2. Materials and Methods

### 2.1. Design

This study will use a case–control design (1 case: 1 control) on 150 mother–child pairs. Cases will be mother–child pairs in which the child has suspected enamel anomalies on eligible teeth, and controls will be children with no suspected enamel anomalies on eligible teeth (based on a clinical exam). During the study, one exfoliated or extracted tooth will be collected from each child and analyzed, and the mother will be interviewed to learn about the child’s adverse exposures, access to support and other family information. Logistic analyses will assess the relationship between the child’s adverse exposures and access to supports and presence of enamel anomalies using different assessment modalities.

### 2.2. Sample Selection

#### 2.2.1. Participants Will Be Recruited from the Practice of a Pediatric Dentist Located in Central Toronto Canada with a Diverse Patient Population. Study Inclusion and Exclusion Criteria for Mother-Child Pairs

Eligibility will be determined based on a review of the child’s dental chart and the dentist’s assessment of the child’s dental health during their appointment.

Inclusion criteria are:

The child is 5–13 years old on the date of agreement to participate in the study

The mother is the biological mother of the child (so information can be obtained about the pregnancy).

The mother is able to provide consent

The mother can be interviewed in English (as project resources preclude conducting the interviews in multiple languages).

The 5–13 years age range was selected because a child has normally lost their 20 primary teeth by ages 12 or 13, and the study is looking for exfoliated primary molars and primary cuspids [51].

Exclusion criteria

A child has a sibling (or another child living in the same household) already enrolled in the study.

#### 2.2.2. Case and Control Group Eligibility

The case group will include children with suspected enamel anomalies identified during the child’s dental visit by standardized visual assessment [52,53] (see details below). Enamel anomalies (developmental opacities) come in several forms, for example, hypomineralization, hypoplasia, or amelogenesis imperfecta. The most common type is molar-incisor hypomineralization, which is abnormal mineralization of enamel during tooth development [54].

Eligible teeth will be primary molars or primary cuspids or any permanent teeth. The enamel in these teeth is less developed at birth than the enamel in other primary (incisor) teeth [27,55] and is more likely to be affected by postnatal stress exposures [56]. To be eligible, teeth also must have no restorations on the surface with the suspected enamel anomalies at point of care.

The control group will include children with no suspected enamel anomalies at the point of care [53]. Controls will be matched to cases based on age (+1 or −1 year at time of appointment) and child sex.

#### 2.2.3. Consent and Assent

Families using the dental practice who have agreed to be contacted about research studies and who are eligible for the study will be invited to contact a member of the research study team to learn more about the study.

The research team will explain the study and obtain formal consent from interested mothers. This will include explaining the limits to confidentiality and the team duty to report to the Children’s Aid Society if the child is perceived to need protection.

Children who are 10 years or over will also be given a child-friendly information pamphlet and asked to provide assent. There is no established age of assent for research, as it depends on the child’s capacity, and if the child can understand a simple explanation of the research project. The Hospital for Sick Children (Toronto) guidelines often use 10 years+ as the age at which assent must be offered, so the same threshold will be used in this work.

Mother–child pairs may receive up to CAD 65 for participation in all components of the study. Specifically, for signing up for the study, each mother-child pair will receive CAD 15. For interview participation, each mother–child pair will receive an additional CAD 35 honorarium to cover childcare and time costs, and an additional CAD 15 for giving a tooth to the study team.

### 2.3. Recruitment and Sample Size

Recruitment into the study will occur over a one-year period. Individuals will be assigned to the case or control groups sequentially until both case and control groups are filled. To achieve a final sample size of 150 (with equal cases and controls), we will recruit 220 mother–child pairs. This is based on the expectation that up to 35 children will either not lose an eligible tooth or not notify the study team during the follow-up period, and that an additional 35 mothers will be lost to follow-up or decline to be interviewed. Recruitment of 220 mother–child pairs is feasible based on an estimated sampling pool of 953 mother–child pairs (estimated assuming that the participating dentist sees an average of 25 patients/day and works 225 days/year. Of them, 70% are unique patients (i.e., not repeat visitors), 95% have English proficiency, and 85% are 5–13-year-olds, and about 30% of children have enamel anomalies) involving unique patients 5–13 years of age seen in the practice annually of the referring dentist and a minimum 24% rate of acceptance.

Due to the scarcity of data on our hypothesis and the exploratory nature of this study, we cannot accurately calculate statistical power. We expect that the sample of 150, with 75 enamel anomalies identified through standardized visual assessment (i.e., outcomes of interest) will provide sufficient power to enable fitting of the data to the convergence of the multivariable model.

### 2.4. Variables and Data Collection

#### 2.4.1. Outcomes

The presence of any enamel anomalies (present, not present) as identified in clinical practice through standardized visual assessment will be the primary outcome. The presence of any enamel anomalies (present, not present) will also be assessed using three other modalities described below (see Secondary Outcomes Section). Identification using visual assessment will be the primary outcome, being part of routine clinical practice. PTR-LUM is also a feasible clinical strategy, but is presently less available as it requires the dental office to purchase additional specialized equipment. The other two methods can only be applied in laboratory settings to extracted or exfoliated (rather than in-mouth) teeth; as such, they are not available for in-clinic use at this time.

#### Primary Outcome

The dentist will apply a standardized method for assessment of enamel anomalies using the index of developmental defects of enamel (modified developmental defects of enamel (DDE)) (Table 1) [52]. This is important as clinical visual inspection practices for enamel anomalies can vary [57]. The Modified DDE Index has been extensively used since its introduction in 1992 and has shown high degrees of validity and reliability [58,59,60,61,62,63] (for more details, see Appendix A). Evidence of dental decay (caries) will be coded as present or not present. Additionally, for descriptive purposes, anomalies will be categorized by type (hypomineralization, hypoplasia, or amelogenesis imperfecta, or other) and differentiated from caries using these published guidelines and definitions (Table 1) and a table that members of our study team developed (Table 2) to augment existing guidelines.

The dentist will also apply the International Caries Detection and Assessment System II (ICDAS-II) [53] to identify if the tooth shows evidence of cavity tooth decay (dental caries). Anomalies will be differentiated from caries using published guidelines and definitions and a table that members of our study team developed (Table 2) to augment existing guidelines.

#### Secondary Outcomes

Each of the below methods will be applied to all extracted or exfoliated teeth to assess and confirm the presence of enamel anomalies (e.g., in cases and controls as determined by visual inspection). Each of these may lead to a regrouping of cases and controls for secondary analyses.

Photothermal Radiometry and Modulated Luminescence (PTR-LUM). PTR-LUM is a laser-based method for oral health assessment performing the detection, measurement and monitoring of changes in tooth structures including enamel. It is used in clinical practice and research and has greater sensitivity and accuracy in detecting and measuring caries and greater potential for detecting enamel anomalies than visual assessment or X-rays [64,65,66,67,68]. In contrast to X-rays, PTR-LUM emits only thermal infrared radiation, which is not harmful in multiple exposures and is more accurate than radiographs for detecting caries [64,65,66,67,68,69]. Studies have shown that it does correlate with microcomputed tomography measurements [70,71]. PTR-LUM is an in-clinic method diagnostic tool that is currently available in some practices.


*Laboratory based modalities—3D imaging and reconstruction technologies*


As with PTR LUM, truncated correlation-photothermal coherence tomography (TC-PCT) is a laser-based assessment method. It is also a dynamic thermography imaging technique that is very sensitive in contrast, which aids in early diagnosis of oral health problems. This emerging technology is expected to be available for in-clinic use during later stages of its development [72,73].

Microcomputed tomography (Micro-CT, or μCT) provides high-resolution images of the internal structure of a tooth, including mineral density, volume (of pores or hard tissue) and depth in teeth and bone [74,75]. This X-ray imaging method has high spatial resolution and has high sensitivity to detect the most minimal defects or very early-stage caries lesions, and as such it is currently used as one of the reference standards for in vitro hard tissue (bone and teeth) studies. In this study, μCT will provide the reference standard for validating the assessment of enamel anomalies.

#### 2.4.2. Child Tooth Collection

One exfoliated or extracted tooth will be collected from each participant during the year following study enrolment. Those in the case group will need to provide the tooth identified with an enamel defect during their initial dental visit. Mothers will receive an image of the tooth of interest and provide a picture of the child’s mouth after the tooth has fallen out to confirm that it is the desired tooth.

For exfoliated teeth, families will put the tooth in water in a study-provided specimen container and notify the dental staff (see Figure 1 for the full process) who will arrange for its transfer to the dental office. To be eligible for the study, the tooth must be received by the dental staff within 10 days of the exfoliation. At the dental office, the tooth will be put into a new container containing 70% ethanol and kept at room temperature for 72 h for disinfection. After 72 h, the staff will transfer the tooth to a new container with neutralized distilled water that is changed weekly and stored in a dedicated refrigerator at the dental practice.

For teeth that are extracted during the clinic visit, the process will be the same, except that the tooth will be immediately transferred to the container with the 70% ethanol solution for 72 h and then moved to the container with neutralized distilled water in the refrigerator for long-term storage.

All teeth will be labeled with the method of tooth loss, the method of extraction if relevant, the tooth number and the type of tooth to account for this in the multivariable analysis. The staff will change the water weekly until all analysis modalities have been completed on that tooth.

#### 2.4.3. Maternal Interview

The interviewer will administer measures to assess the following: child exposure to adversity; child resilience; and other child and family factors that may affect tooth health. The interview will be conducted by trained social workers who are supervised by the Victimology Research Centre at Algonquin College and who will complete the Adverse Childhood Experiences (ACEs) Aware training (https://www.acesaware.org/) (accessed on 19 July 2021) on safe and ethical ACE screening.

#### Adversity Measures (Exposure Variable)

Children’s exposure to adversity will be assessed using the Pediatric ACEs and Related Life Events Screener (PEARLS). The PEARLS is used for screening for childhood adversity by primary care providers in California as part of a new state-wide initiative [76]. The PEARLS [77] includes two sections: deidentified and identified. The “deidentified” section includes the questions from the original ACE questionnaire [2,78], adapted to be more inclusive of gender (i.e., victims of intimate partner violence are not assumed to be exclusively women) and of caregivers beyond parents. Mother respondents will review 10 types of adversity exposures and report the total number experienced by their child to the interviewer without identifying which ACEs were experienced. The “identified” section asks about exposure to adversities not included in the original ACE questionnaire such as experiences with violence, discrimination, food or housing insecurity, separation from a caregiver, or the serious illness or death of a caregiver. For each item in this section, the respondent indicates yes or no. The positive responses are summed to yield a total score out of 7 [77].

Children’s exposure to neighborhood-level adversities will be assessed using items adapted from Mountain et al. [31].

#### Resilience Measures (Modifying Variable)

Resilience will be measured via the seven-item positive childhood experiences (PCEs) scale and three items from the 10-item benevolent childhood experiences (BCEs) [79] scale that do not overlap with the PCE scale. Both PCEs and BCEs [15,17] capture experiences before age 18 that are thought to promote wellness and be beneficial to a child, such as positive relationships with parents and other adults, household routines, beliefs that provide comfort, and having good neighbors [14,15,16,17]. The mother will rate each item as present or not. Positive responses will be summed to create a total resilience score out of 10.

#### Other Maternal and Family Information (Descriptors)


Information on other factors related to the development of enamel anomalies will also be collected. These include:Prenatal and perinatal factors: prenatal urinary infections [80], pregnancy parity, child gestational age at delivery, mode of delivery [81], use of breast and/or bottle feedingLifestyle factors: child dietary habits, access to regular dental care, oral hygiene [56], and frequency of drinking fluoridated tap waterChild health illnesses and use of antibioticsFamily information: family structure, parent ethnicity, parent education and employment, annual household income


PCEs were adapted from four subscales in the Child and Youth Resilience Measure—the Psychological, Caregiving subscale; the Education subscale; the Culture subscale; and the Peer Support subscale.

### 2.5. Data Analysis

Univariate analyses will compare case and control groups on all descriptor variables (adversity exposures, resilience modifier, and other maternal and family information). For continuous variables or ordinal variables, paired *t*-tests or a Wilcoxon sum rank test will be used, depending on the distribution. For nominal variables, chi-square tests or a Fisher’s exact tests will be performed. Continuous variables will be expressed as a mean ± standard deviation or as a median with the interquartile range, depending on the distribution. Categorical (nominal or ordinal) variables will be summarized as frequencies and percentages.

Multivariate analysis: To assess the association between the presence of an enamel anomaly on the child’s tooth and the child’s total adversity scores, conditional logistic regression modeling will be conducted. In the main model, the outcome will be the presence of enamel anomalies identified at the point of care (case versus control). Predictors will be: the ACE questionnaire score, the PEARLS “deidentified” section score and the neighborhood exposure to adversities score. The results will be reported as the odds ratio between the case and control groups for enamel anomalies identified at point of care—present/absent, with a 95% confidence interval. Additional regression models will assess the association between the three adversity scores and the presence of enamel anomalies assessed using the other three modalities. Multivariable models of secondary outcomes will account for the selection bias in the case–control selection process by implementing strategies suggested in frameworks for these analyses in case–control studies [82,83].

We will also fit the four models with resilience scores (out of 10, from summing positive responses to questions on access to resilience building resources questions [9]) as predictors to assess whether the impact of childhood adversity on the presence of anomalies (tested in separate models) is related to the child’s access to resilience building resources.

All analyses will be performed using R software, version 4 [84].

### 2.6. Ethics

Ethical committee approval is being requested from the Research Ethics Boards of the participating institutions including University of Toronto, St. Michael’s Hospital, and the Institutional Review Board of the University of Texas Health San Antonio, Texas. Approval from the University of Toronto has already been received.

## 3. Discussion

This proof-of-concept, case–control study will yield the first empirical evidence of the association between structural changes in human teeth enamel and a child’s exposure to adversity. The aim is to identify a dental biomarker of toxic stress exposure that can be assessed during routine clinical care and potentially support referral for early intervention. A small body of studies on children and adults have reported an association between dental problems and childhood adversity [47,48]. However, the most common dental problems (tooth decay) develop in response to lifestyle factors that occur post-tooth eruption (e.g., sugary foods consumption, oral hygiene, exposure to fluoride in toothpaste or water sources, and/or access to professional dental care). As such, dental problems broadly defined have low sensitivity for indicating exposure to the toxic stress linked to childhood adversity. A more specific study [85] showed an association between adverse events (using traditional 10-item ACE questionnaire) and early molar eruption as assessed with magnetic resonance images. While this study suggests that toxic stress from exposure to childhood adversity is linked more specifically to molar disruption, this is not a biomarker that can be assessed in routine clinical dentistry. By focusing in the present study on an outcome that is detectable in routine practice, we hope to advance efforts to create a role for dentistry in the early identification of children’s exposure to adversity.

While there may be opposition initially to expanding the duties of OHCPs, the notion that dentists should contribute to a patient’s overall health is not new. It has been shown that screening of elevated risks for particular diseases (e.g., diabetes, cardiovascular disease) during dental appointments by OHCPs supports subsequent linking with the medical system for diagnosis or risk monitoring. Conducting screening in dental settings for medical conditions has been viewed favorably by OHCPs, dental patients, and primary care physicians [86,87,88]. Expanding the scope of practice for OHCPs to screen for flags that a patient may be in the early stages of experiencing physical health issues aligns with recent claims that dentistry holds a potentially important role in facilitating change and in supporting greater equity in health. One way in which dentistry could support greater equity is by helping to identify childhood adversity [89].

There are two key ethical issues that must be considered in the context of this work. The first is resistance to a universal “screening tool” for childhood adversity because of the potential for revictimization, increased stigma from providers [90,91], and because comprehensive trauma and violence-informed approaches are often scarcely available [91,92,93,94,95]. Even so, several jurisdictions (e.g., California) are seeking to implement universal screening among children and adults in primary care settings for childhood adversities and to treat the impacts of toxic stress on stress-related physical and mental health concerns with trauma-informed care and evidence-based interventions [96]. Another key issue is that although the role of biomarkers in early intervention delivery is growing [97], biomarkers have complex histories of being used to perpetuate racism [97], and can be difficult to disentangle from social conditions of inequality that produce poorer health outcomes for different populations. Still, biomarkers are used in medical practice to guide prevention and early intervention, for example, to predict patient conversion from clinical high risk to frank psychosis, to aid in detection of risk for dementia, and to predict kidney disease [97]. In addition, it will be critical for future work to carefully analyze the risks and benefits of biomarkers for each clinical application [97]. Related to the present application, health professionals need to be trained in sensitive conduct screening for adversity, without traumatizing or re-traumatizing the child and family, and in not drawing faulty assumptions about a child’s future [98].

### 3.1. Limitations

This study has several limitations. First, mothers tend to underreport adversities experienced by their child due to several factors, including a fear of social stigma [27,99]. Still, parent-report is the most widely accepted way of measuring children’s exposure to adversities [85,100]. The interviewer will be trained to assess early childhood trauma as well as current risk to self and others and will adopt the approach used by the ACEs Aware program in California. Second, because primary teeth begin to develop while a child is in utero and continue after birth [96], it is possible that enamel anomalies may reflect experiences that occurred in utero. To minimize this issue, only primary teeth that develop at later stages of pregnancy are eligible for this study. Third, caution will be required when extrapolating results beyond the study sample, as it may not be representative of the general dental clinic visiting population. This includes ethno-racial and socioeconomic diversity. Fourth, access to dental care is not universal, although a national Canadian survey reported that 86.9% of 12–17-year-olds had visited a dentist in the past 12 months [101]. Fifth, the study team had to select the assessment modalities to include in the study based on expectations of success, accessibility of the equipment, expertise of the study team, and cost. In future work, it would also be desirable to include other modalities, such as transverse microradiography (TMR).

### 3.2. Study Strengths

This exploratory, proof-of-concept study is a first step to investigating a potentially expanded role for OHCPs in screening dental patients for exposure to childhood adversity. As noted earlier, childhood adversity has been labeled one of “society’s most complex and enduring problems” [102]. OHCPs are well-positioned to take on a screening and referral role. They are a familiar and trusted health professional. They commonly develop long-term relationships with families and may see children who do not regularly access general medical care. In the US, 26% of children reportedly do not interact with general healthcare providers in a year, but of them, 34.7% were seen by a dental professional at least once during that same year [50]. Providing OHCPs with a potential biomarker can provide an opportunity for discussion and timely referral, and expand upon their current common practice of screening for physical health conditions such as diabetes and heart disease [103,104].

Methodologically, this work is strengthened by using multiple assessment modalities with varying sensitivity levels to identify biomarkers, including those that can be incorporated into routine care, and reference standard modalities. It is also strengthened by having a study team from disciplines that do not traditionally collaborate.

## 4. Conclusions

This research seeks to identify a dental biomarker of childhood adversity exposure to target early intervention. The presence of a biomarker will be assessed using multiple modalities, including those that can be incorporated into routine care. Using a case–control design, the study will assess a child’s exposure to adversity and resource building resources as well as measures of oral health. Such studies provide a first step in determining if routine dental exams can play a role in identifying the non-dental challenges children are experiencing and link them to supports if desired.

If a relationship is observed between tooth enamel anomalies and childhood adversity, the next steps will be to replicate the study with larger and more diversified samples, and to ask dentists what additional resources they need to contribute to screening for children with possible exposure to adversity. How the observed findings could be shared with dental practices will also be examined.

## Figures and Tables

**Figure 1 ijerph-19-03403-f001:**
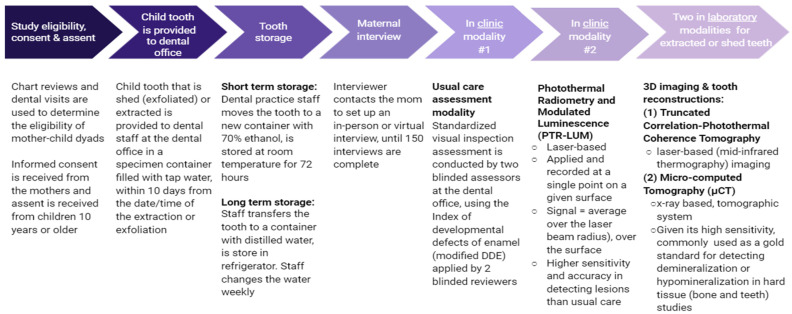
Study processes.

**Table 1 ijerph-19-03403-t001:** Scoring system of the Modified Developmental Defects of Enamel (DDE) Index for epidemiological studies [58].

**Type of Defect**	**Code**
Normal	0
Demarcated opacities (smooth surface without discontinuity):	
White/cream	1
Yellow/brown	2
Diffuse opacities (smooth surface without discontinuity):	
Diffuse—lines	3
Diffuse—patchy	4
Diffuse—confluent	5
Confluent/patchy + staining + loss of enamel	6
Hypoplasia (deficiency in amount of enamel development, i.e., there is discontinuity):	
Pits, fissures, grooves or furrows	7
Missing enamel	8
Any other defect	9
**Severity of anomaly (Extend of defect)**	**Code**
Normal	0
<1/3 of the surface	1
At least 1/3 and <2/3	2
At least 2/3	3

**Table 2 ijerph-19-03403-t002:** Differentiating the causes of white spot lesions as caries versus enamel anomalies.

Criteria for Distinction	Caries	Enamel Anomalies
Appearance	Opaque, chalky and dull (matt) surface when air-dried	Glossy surface when air-dried
Texture	Feels rough when the tip of the explorer is moved gently across surface	Feels smooth when the tip of the explorer is moved gently across the surface
Shape	Elliptical or crescent shaped	Lines resembling pencil shading
Area affected	Located in plaque stagnation areas (gingival 1/3, pits/fissures, proximal surfaces)	Located mainly in self-cleansing areas (incisal edges, cups tips, occlusal 1/3 and center of smooth surfaces, may affect entire crown)
Distribution	May affect a single tooth (gingival 1/3, pits/fissures, proximal surfaces)	Multiple teeth involvement (i.e., bilateral or quadrilateral on corresponding (sister) teeth in the same location and with the same shape)

## Data Availability

Not applicable.

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
