# Peer review of "Protocol for a Case Control Study to Evaluate Oral Health as a Biomarker of Child Exposure to Adverse Psychosocial Experiences"

_ijerph, 2022, doi:10.3390/ijerph19063403_

Round 1
Reviewer 1 Report
- Please add the approval number for human trials.
- I didn't find a clear number of subjects.
- The results of this study must be added. 2. Materials and Methods, 3. Results and 4. Discussion
- 2. Materials and Methods
- 2. The narration of Materials and Methods is very complicated and needs to be rewritten.
- What does the article refer to as a biomarker?
- Biomarkers can refer to many types. In the abstract, I think it should be precise to explain what is used as a biomarker.
- In addition, the manuscript mentions that there are human studies, but how the calculation results and analysis are not shown. Contains statistical analysis.
- There are too many References, and important documents should be kept.
Author Response
Thank you for the opportunity to revise our manuscript. Our responses to reviewer comments are below in bold font.
Please add the approval number for human trials.
This is not a trial, as such there is no approval number
I didn't find a clear number of subjects.
As this a protocol paper we indicate the planned (not achieved) number of subjects. We are aiming for a final sample of 150 mother-child pairs. To achieve this, we will recruit 220 mother-child pairs into study as attrition is expected (see study design page 5 and sample recruitment page 7).
The results of this study must be added. 2. Materials and Methods, 3. Results and 4. Discussion
As this is a protocol paper there are no study results yet to report.
The narration of Materials and Methods is very complicated and needs to be rewritten
We added a paragraph that contains a brief overview of the study in section 2.1.
What does the article refer to as a biomarker? Biomarkers can refer to many types. In the abstract, I think it should be precise to explain what is used as a biomarker.
We have added text to the abstract and the introduction to better explain what a biomarker is for the purposes of this study.
In addition, the manuscript mentions that there are human studies, but how the calculation results and analysis are not shown. Contains statistical analysis.
We have added more detail on the methods used in these studies, as well as the results. We also added details on a recently published study that examined the width of neonatal lines at three sections in the canine teeth of 70 children aged 5-7 years old to assess if width of these neonatal lines were related to prenatal and perinatal maternal psychosocial factors.
There are too many References, and important documents should be kept.
Our aim was to provide clear support for the rationale, aims and methods of the study. Many of the references are for the technical descriptions of the assessment modalities. However, we have eliminated a small number of references to avoid overburdening the reader.
Reviewer 2 Report
This is a well written protocol for a case control study to investigate the association between dental anomalies and exposure to adversity. I reviewed this paper in conjuction with the STROBE Statement—Checklist of items that should be included in reports of case-control studies (https://www.strobe-statement.org/download/strobe-checklist-case-control-studies-pdf) and found it comprehensive.
I have some minor comments that I believe would enhance section 2.5 Data Analysis.
Please state explicitly what result will you report from the logisitic regression and include a precision estimate for the result. For example: The results will be reported as the odds ratio between the two groups (enamel anomaly present/ absent) with a 95% CI.
There is no need to report p values; reference: Amrhein, Valentin, Sander Greenland, and Blake McShane. "Scientists rise up against statistical significance." (2019): 305-307)
Please include information on how missing data (including where a response of "prefer not to say" is recorded) will be managed.
Bias is mentioned with regard to the secondary analyses and potential reclassification of case and control group allocation, but there is no comment on how the presence of bias in the current study would be checked and managed. For example how would a potential imbalance between the groups with regard to factors listed in 2.4.3.3. Other maternal and family information (descriptors) - be managed? There is evidence that parent education is associated with oral health (outcome) and adversity (exposure). I strongly encourage the authors to consider creating a Directed Acyclic graph (DAG) to consider potential confounding and intermediary variables.
Author Response
Thank you for the opportunity to revise our manuscript. Our responses to reviewer comments are below in bold font.
This is a well written protocol for a case control study to investigate the association between dental anomalies and exposure to adversity. I reviewed this paper in conjuction with the STROBE Statement—Checklist of items that should be included in reports of case-control studies (https://www.strobe-statement.org/download/strobe-checklist-case-control-studies-pdf) and found it comprehensive.
I have some minor comments that I believe would enhance section 2.5 Data Analysis.
Please state explicitly what result will you report from the logisitic regression and include a precision estimate for the result. For example: The results will be reported as the odds ratio between the two groups (enamel anomaly present/ absent) with a 95% CI.
We have added the sentence: “The results will be reported as the odds ratio between the case and control groups for enamel anomalies identified at point of care - present / absent, with a 95% confidence interval.”
There is no need to report p values; reference: Amrhein, Valentin, Sander Greenland, and Blake McShane. "Scientists rise up against statistical significance." (2019): 305-307)
Thank you. We removed: “A two-tailed p-value of ≤0.05 will be considered statistically significant.”
Please include information on how missing data (including where a response of "prefer not to say" is recorded) will be managed.
Thank you for flagging this issue. Per variable, we will use imputation or exclude participants as needed if there are very high levels of missing values.
Bias is mentioned with regard to the secondary analyses and potential reclassification of case and control group allocation, but there is no comment on how the presence of bias in the current study would be checked and managed. For example how would a potential imbalance between the groups with regard to factors listed in 2.4.3.3. Other maternal and family information (descriptors) - be managed? There is evidence that parent education is associated with oral health (outcome) and adversity (exposure). I strongly encourage the authors to consider creating a Directed Acyclic graph (DAG) to consider potential confounding and intermediary variables.
We have created this diagram, see attached. Please confirm if it should be incorporated into the paper or included in the supplement.
Although the cases and controls may differ in the prevalence of key variables (e.g. income), we will assess differences across groups and include these variables as covariates in the multivariable regression model where appropriate. Of course, residual confounding is always a risk.

Reviewer 3 Report
Dear authors,
I read your proposal with interest, I express positive judgment about continuing its editorial process not before having clarified some minor issues:
- About micro-TC: ethical considerations must be posed about this element. Later on in your manuscript you declared that micro-TC will be considered the gold standard in your study. But there is difference between considering something the gold standard in a study and the gold standard in general as reported by scientific literature. In this second case ethical considerations could be a minor issue but otherwise this aspect should be discussed appropriately.
- Will Ethical Committee approval be requested? The protocol only report a generic "guidelines will be followed".
- Also children with no suspected enamel anomalies will be recruited as dyads?
Regards
Author Response
Thank you for the opportunity to revise our manuscript. Our responses to reviewer comments are below in bold font.
REVIEWER COMMENTS
I read your proposal with interest, I express positive judgment about continuing its editorial process not before having clarified some minor issues:
About micro-TC: ethical considerations must be posed about this element. Later on in your manuscript you declared that micro-TC will be considered the gold standard in your study. But there is difference between considering something the gold standard in a study and the gold standard in general as reported by scientific literature. In this second case ethical considerations could be a minor issue but otherwise this aspect should be discussed appropriately.
We have changed the language in the manuscript to be “reference standard” rather than “gold standard”. Still, there is no single method that serves as reference standard for measurement of all outcomes as it depends on the outcomes to be measured. For example, the reference standard for confirming the presence and depth of hypomineralization in a tooth is histological method and at times transverse microradiology; however, these methods are destructive [1,2], cannot quantify the mineral density or the volume of the hypomineralized tissue, and their measurements are limited to only a thin slice from the defect under examination. MicroCT allows examination of teeth without destruction of the samples [3,4], can measure the mineral density and the volume of the entire hymineralized tissue [83,84], and produces 3-D image of the tooth that permits the measurement of the entire hypomineralized defect. These being our outcomes of interest, microCT was then chosen as the reference standard in this study. Furthermore, microCT has been used in many dental research publications over the last number of years [5,6], and it provides lab findings that support clinical studies related to caries and enamel and dentin structure [7].
References:
- Ricketts DN, Watson TF, Liepins PJ, Kidd EA. A comparison of two histological validating techniques for occlusal caries. J Dent. 1998;26:89–96. [PubMed] [Google Scholar]
- Jablonski-Momeni A, Stachniss V, Ricketts DN, Heinzel-Gutenbrunner M, Pieper K. Reproducibility and accuracy of the ICDAS-II for detection of occlusal caries in vitro. Caries Res. 2008;42:79–87. [PubMed] [Google Scholar]
- Mitropoulos P, Rahiotis C, Stamatakis H, Kakaboura A. Diagnostic performance of the visual caries classification system ICDAS II versus radiography and micro-computed tomography for proximal caries detection: An in vitrostudy. J Dent. 2010;38:859–67. [PubMed] [Google Scholar]
- Huang TT, Jones AS, He LH, Darendeliler MA, Swain MV. Characterisation of enamel white spot lesions using X-ray micro-tomography. J Dent. 2007;35:737–43. [PubMed] [Google Scholar]
- Hamba H, Nikaido T, Sadr A, Nakashima S, Tagami J. Enamel lesion parameter correlations between polychromatic micro-CT and TMR. J Dent Res. 2012;91:586–91. [PubMed] [Google Scholar]
- Borba M, Miranda WG WG, Jr, Cesar PF, Griggs JA, Bona AD. Evaluation of the adaptation of zirconia-based fixed partial dentures using micro-CT technology. Braz Oral Res. 2013;27:396–402. [PubMed] [Google Scholar]
- Özkan G, Kanli A, BaÅŸeren NM, Arslan U, Tatar Ä°. Validation of micro-computed tomography for occlusal caries detection: An in vitrostudy. Braz Oral Res. 2015;29:S1806–83242015000100309. [PubMed] [Google Scholar]
Will Ethical Committee approval be requested? The protocol only report a generic "guidelines will be followed".
We have clarified this text to explain that we are seeking approval from all of the organizations participating in the study. The sentence now reads, “Ethical Committee approval is being requested from the Research Ethics Boards of the participating institutions including University of Toronto, St. Michael’s Hospital, and the Institutional Review Board of the University of Texas Health San Antonio, Texas. Approval from the University of Toronto has already been received.“
Also children with no suspected enamel anomalies will be recruited as dyads?
The term ‘dyads’ is used in this paper to refer mother-child pairs. We have removed the term “dyads” and now use “mother-child pairs.” We will also match cases to controls.
Reviewer 4 Report
Dear authors,
I had the opportunity of revising the present case-control study protocol regarding if oral health can provide early indicators of child exposure to adverse psychosocial experiences.
The protocol is interesting and very well written but I feel without a complete study with full data the publication of IJERPH is not recommended. I will be very happy of revising the manuscript after presentation of the complete data set analysis and discussion.
Best Regards
Author Response
Thank you for the opportunity to revise our manuscript. Our responses to reviewer comments are below in bold font.
REVIEWER COMMENTS
I had the opportunity of revising the present case-control study protocol regarding if oral health can provide early indicators of child exposure to adverse psychosocial experiences.
The protocol is interesting and very well written but I feel without a complete study with full data the publication of IJERPH is not recommended. I will be very happy of revising the manuscript after presentation of the complete data set analysis and discussion.
Thank you for your interest in our study. We are submitting this protocol paper as interest in role of teeth as biomarkers of psychosocial stressors is growing and it is important to share our study aim and methods with interested colleagues. We will certainly write a paper to share our findings with broad audiences when they are available.
Reviewer 5 Report
Overall I found this an interesting study protocol with important potential findings. I would suggest a number of changes to address typographical and readability issues. These include, but are not limited to:
Line 34 add "y" to "laborator"
Line 37 consider revising sentence so as not to start with "then".
Line 87 consider revising sentence to replace "sorely needed". Perhaps just delete "sorely".
Line 96 consider replacing "search' with "need" or similar.
Line 100 I'd consider rewording the phrase "identifiable with limited invasiveness" to improve readability.
Line 125 Missing full stop.
Lines 126-127 I'd consider rewording the phrase "With usual care of visual assessment ..." to improve readability.
Line 157 Delete “)”
Line 161 consider revising sentence to start with "In summary".
Line 170 Delete “).”
Line 183 consider revising sentence to finish with "..as a proof of concept" instead of “(proof of concept).
Lines 194-197 It would have been good to see in the methods an indication of the likely SES/racial mix of the practice. Is it broadly representative of the wider local/regional population? Are you likely to get a spread of ACE/social disadvantage?
Line 245 consider revising sentence to finish with “and the child’s sex.”
Line 295 fix formatting of “visual”
Line 313 consider using “dental decay (caries)” instead of “cavity decay (caries)”
Lines 348-349 Is this meant to be a sub-heading? It looks out of place as a sentence and doesn’t have a full stop.
Lines 436-449 It would be good to see a bit more detail on how confounding will be managed from SES factors other than those directly related to ACE. Also, will there be any enamel defects related to fluorosis that may be an issue? Does the area have water fluoridation?
Lines 599-600 It would be good to consider how the dental/oral health team may respond to developing new service models in collaboration with primary health care service providers, rather than just referral pathways. What will happen to referred children and is there a need to think more about new solutions – can dentists and oral health therapists be part of a wider solution? This is not a criticism, but we need to think of what will be do in response to this information that may help these children and families. This is of course why this research is important.
Author Response
Thank you for the opportunity to revise our manuscript. Our responses to reviewer comments are below in bold font.
REVIEWER COMMENTS
Overall, I found this an interesting study protocol with important potential findings. I would suggest a number of changes to address typographical and readability issues. These include, but are not limited to:
Line 34 add "y" to "laborator" Added
Line 37 consider revising sentence so as not to start with "then". We have removed “then”. It now states that “If this association is present oral health care providers could contribute to early screening for children's exposure to adversity and referral to appropriate follow up services.
Line 87 consider revising sentence to replace "sorely needed". Perhaps just delete "sorely". Removed “sorely”
Line 96 consider replacing "search' with "need" or similar. “Search” has been replaced with “Need”
Line 100 I'd consider rewording the phrase "identifiable with limited invasiveness" to improve readability. Changed to “However, they need to be identifiable with limited cost and minimally invasive approaches”
Line 125 Missing full stop. Added a period
Lines 126-127 I'd consider rewording the phrase "With usual care of visual assessment ..." to improve readability. It is now rephrased as “During clinical dental exams that routinely use visual assessment (usual care), there is often low precision of identification of enamel anomalies. Diagnosis”
Line 157 Delete “)” Removed
Line 161 consider revising sentence to start with "In summary". Done
Line 170 Delete “).” Removed
Line 183 consider revising sentence to finish with "..as a proof of concept" instead of “(proof of concept). Revised
Lines 194-197 It would have been good to see in the methods an indication of the likely SES/racial mix of the practice. Is it broadly representative of the wider local/regional population? Are you likely to get a spread of ACE/social disadvantage? Patients who are participants in the Healthy Smiles Ontario (HSO) program represent 23% of the practice, which is a proxy for some form of disadvantage. HSO is a provincial program, implemented in 2016, that provides no-cost dental services to children in low-income families earning between $22,000-$38,000, to children in families that receive government social assistance (e.g., Ontario Works) or to children. that received disability support (e.g., Ontario Disability Support Program (“Healthy Smiles Ontario (HSO) - Health n’ Smiles, https://www.smilecaredental.ca/blog.)
We have added a settings paragraph to the method section that more briefly describes the information above.
Line 245 consider revising sentence to finish with “and the child’s sex.” Revised
Line 295 fix formatting of “visual” We believe that we addressed the formatting problem with the word “visual”. If the issue persists, please advise.
Line 313 consider using “dental decay (caries)” instead of “cavity decay (caries)” Replaced cavity decay with dental decay
Lines 348-349 Is this meant to be a sub-heading? It looks out of place as a sentence and doesn’t have a full stop. Yes, the formatting has been changed to be more clear.
Lines 436-449 It would be good to see a bit more detail on how confounding will be managed from SES factors other than those directly related to ACE. Reviewer 2 also raised this issue. We will conduct descriptive analyses to identify key differences across groups and control for them as sample size allows in the multivariable regression model. Of course, residual confounding is always a risk.
Also, will there be any enamel defects related to fluorosis that may be an issue? Other enamel defects (that are not carious cavitation) such as fluorotic defects will be classified under Modified DEVELOPMENT DEFECTS OF ENAMEL (DDE) Index score 9. Fluorosis does not lead to cavitation but Molar Hypomineralisation can lead to cavitation”.
Does the area have water fluoridation? Thank you for raising the fluorosis issue. The entire city of Toronto has water fluoridation, as we now indicate in the settings section of the methods. In the maternal interview, we also ask about if participants have lived outside in Toronto at other points in their lives. We now explicitly note that we will inquire about if child regularly drinks tap water in the maternal interview
Lines 599-600 It would be good to consider how the dental/oral health team may respond to developing new service models in collaboration with primary health care service providers, rather than just referral pathways. What will happen to referred children and is there a need to think more about new solutions – can dentists and oral health therapists be part of a wider solution? This is not a criticism, but we need to think of what will be do in response to this information that may help these children and families. This is of course why this research is important.
We agree. There have bee recent assertions that the dental profession is failing to adequately address population needs and inequities in oral health [1], it is also recognized that dentistry holds a potentially important role in facilitating change and in supporting greater equity in oral health.
Reference
[1] Moeller J, Quiñonez CR. Dentistry's social contract is at risk. J Am Dent Assoc. 2020 May;151(5):334-339. doi: 10.1016/j.adaj.2020.01.022. PMID: 32336345.
Round 2
Reviewer 1 Report
The content has been moderately adjusted.
It is suggested that "Protocol" can be emphasized in the manuscript.
Please reduce the content of the abstract.
Author Response
Thank you for the opportunity to revise our manuscript. Our responses to reviewer comments are below in bold font.
The content has been moderately adjusted.
It is suggested that "Protocol" can be emphasized in the manuscript.
We agree with this suggestion and have added more descriptions of this as a protocol paper.
Please reduce the content of the abstract.
We restructured and cut down the abstract to 189 words.
Reviewer 4 Report
As already told the protocol is very well written.
I just suggest to shorten the introduction that is now too long.
Did you perform any power analysis to identify the required sample size of 150 cases?
Best Regards
Author Response
Thank you for the opportunity to revise and resubmit our paper. Our responses are below in green.
As already told the protocol is very well written.
Thank you.
I just suggest to shorten the introduction that is now too long.
We agree that it had become too long and have reduced the word count in the introduction.
Did you perform any power analysis to identify the required sample size of 150 cases?
We have already have written the following in the paper and do not know what to add : "Due to the scarcity of data on our hypothesis and the exploratory nature of this study, we cannot accurately calculate statistical power. We expect the sample of 150, with 75 enamel anomalies identified through standardized visual assessment (i.e., outcomes of interest) will provide sufficient power to enable fitting of the data to the convergence of the multivariable model"